# Examining the Nonlinear and Synergistic Effects of Multidimensional Elements on Commuting Carbon Emissions: A Case Study in Wuhan, China

**DOI:** 10.3390/ijerph20021616

**Published:** 2023-01-16

**Authors:** Liang Guo, Shuo Yang, Qinghao Zhang, Leyu Zhou, Hui He

**Affiliations:** 1School of Architecture & Urban Planning, Huazhong University of Science and Technology, Wuhan 430000, China; 2The Key Laboratory of Urban Simulation for Ministry of Natural Resources, Wuhan 430000, China

**Keywords:** nonlinear and synergistic effects, multidimensional elements, commuting carbon emissions

## Abstract

Understanding the specific effects of multidimensional elements of a built environment, transportation management policies, and the socio-demographics of travelers associated with commuting carbon emissions is significant for planners in promoting low-carbon and healthy urban development through transportation and land use and urban management policies. Most of the existing studies focus on the complex mechanisms affecting commuting behavior, but the relevant elements and specific mechanisms affecting commuting carbon emissions have not received sufficient attention. This study uses a random forest approach to analyze residential travel data from Wuhan, China. The results show that built environment and transportation demand management policies are critical to commuting carbon emissions, and that there is a non-linear association between multidimensional factors and commuting carbon emissions in Chinese cities. In addition, the study examines the synergistic effects of built environment and transportation management policies on commuting carbon emissions among different built environment elements. The results of the study provide valuable insights for planners in formulating low-carbon city and transportation development policies.

## 1. Introduction

Transportation-related carbon emissions are major contributors to global climate change. The transportation sector is now the second-largest source of carbon emissions worldwide, accounting for 25% of total emissions [1]. At almost 50% of all trips, commuting is the most important transport activity in daily life [2], and commuting CO_2_ emissions are, therefore, a significant source of transport CO_2_ emissions. Understanding the mechanism by which multidimensional urban factors influence commuting CO_2_ emissions is critical for developing low-carbon transportation and urban development policies and for guiding the healthy development of cities [3]. In the context of China’s goal to achieve carbon peaking by 2030 and carbon neutrality by 2060, transportation carbon reduction is under tremendous pressure, and there is a lack of localized empirical research guidance. Therefore, this study integrates the correlation of multidimensional elements of the built environment, urban transportation management policies, individual socio-demographic factors, and commuting CO_2_ emissions to provide guidance for the healthy development of low-carbon cities.

Existing studies have used both aggregate and disaggregate analyses to analyze the factors and mechanisms that influence travel carbon emissions [4]. Over the last 30 years, several aggregate studies have been conducted on urban form [5,6,7,8], land use [9,10], transport provision [11], transport demand, transport energy use, and carbon emissions, and several valuable conclusions have been drawn. Although they all show that the built environment is significantly related to CO_2_ emissions from travel, a number of issues remain. Overall, aggregate research cannot reveal the mechanism of the built environment’s impact on CO_2_ emissions from individual travel. To address this issue, a portion of the studies used individual (or household) data (disaggregate data) and a disaggregate model [12,13] to investigate the causal pathways of the impact of built environment characteristics on traffic carbon emissions and demonstrated that there is a mediating effect of travel decisions between the two variables.

In recent years, increasing studies have focused on the built environment and travel behavior. Multiple studies have examined the mechanism by which the built environment influences travel behavior [14] and the relative importance of different built environment characteristics regarding travel behavior [15,16,17]. The research framework of “aggregate analysis—comparative study—correlation conclusion” and “disaggregate analysis—mechanism study—causal path inference” has been developed [18]. However, the influencing mechanism of travel CO_2_ still needs further exploration. This is because transport-related carbon emissions are the result of a combination of modal split, the distance travelled by different modes of transport, and mode-specific emission factors. Therefore, built environment variables that affect mode choice and VMT are expected to contribute to the variation in carbon emissions. For example, access to transit facilitates individual transit use and hence reduces carbon emissions. Higher employment density and mixed land use enable residents to internalize their trips within the neighborhoods and promote low-carbon travel behavior [17]. It should be noted that the research base and theoretical basis for the study of travel-related carbon emissions are derived from travel behavior studies. However, as carbon emissions are the result of a combination of travel behaviors, the complexity of this combination may lead to differences in the mechanisms of impact, and, therefore, travel carbon emissions deserve further exploration based on the original research framework. Many studies have touched on the relative importance of different built environment characteristics on travel behavior, but the influence on carbon emissions is in need of more exploration [3,12,13,19]. This is because the current study shows that the key determinants of carbon emissions and travel behavior (mainly mode choice and VMT) somewhat differ [19]. So, we cannot study the factors that affect VMT or the factors that affect the travel mode in isolation and then use the results of these studies as a direct guide to reducing carbon emissions.

In addition, most existing studies assume that the relationship between the built environment and CO_2_ emissions from travel is linear or logarithmic [20,21], which some academics believe is incorrect [22,23]. The latest study is based on a small number of communities in the central areas of American cities and attempts to investigate the nonlinear relationship between them [21]; however, the smaller research scope and small number of samples may result in model over-fitting [24], affecting the accuracy of the results. Furthermore, the form, built environment, provision of facilities, and travel preferences and characteristics of residents are very different between China and Western countries, so the experience of Western countries may not be applicable to China [25,26].

Alternatively, most existing studies focus on the effect of a single built environment factor on CO_2_ emissions, whereas only a few studies discuss the synergistic effect of different factors on CO_2_ emissions when discussing the relative importance of different built environment factors regarding CO_2_ emissions. A nonlinear association implies that the variables have different threshold effects, while synergistic effects indicate the joint effect of multiple variables [27]. This joint effect exists between different dimensional variables, such as the joint effect of TDM and the built environment on travel carbon emissions [23]. It also exists between variables of the same dimension, such as different built environment variables. For example, established studies have shown that the effects of travel cost on the choice of driving alone and non-motorized modes are larger in areas that have a higher sidewalk density and are close to activity centers [28]. This implies a synergistic effect of two built environment variables: sidewalk density and location. This synergistic effect is crucial for urban planners, because once certain factors (such as population density, specific age groups, and urban management policies) have been identified, they have to determine the appropriate range of indicators for other built environment factors. For example, established studies have shown that the combination of land use policy and TDM produces a synergistic effect on travel mode choice [18]. Since land use policies are difficult to change in a short period of time, which TDM policy is the most beneficial for carbon reduction under a given land use indicator? To take another example, in old urban areas, it is difficult to change the land use function due to the difficulty of changing land property rights. If we want to achieve carbon emission reduction through urban renewal, we can only adjust the land use intensity, and it is important to determine the land use intensity value under a specific land use diversity value. Consequently, precisely defining the synergistic effect of various elements and commuting carbon emissions can provide more detailed guidance for specific planning. This study employed a larger sample size to reveal the true relationship between the built environment, traffic demand management, personal socio-demographic characteristics, and carbon emissions, further highlighting the reduction of commuting CO_2_ emissions through urban planning policies.

This study uses resident travel survey data collected in Wuhan, China, a city with typical high-density development characteristics, to examine the nonlinear influence of built environment elements on commuting carbon emissions and the synergistic effect between variables. The goal of this research is to answer the following questions: (1) What are the specific effects and threshold relationships of multidimensional factors influencing commuting carbon emissions? Which dimensions are more important? (2) Do the different variables have synergistic effects on commuting carbon emissions?

The organizational structure of this study is as follows: Section 2 examines the literature on the relationship between the built environment, carbon emissions, and commuting behavior, identifying research gaps. Section 3 introduces data, variables, and the modeling method. Section 4 presents the main results of the study. Section 5 discusses the empirical results of the study and proposes corresponding policy recommendations. The final section summarizes the main conclusions.

## 2. Literature Review

### 2.1. Multidimensional Elements, Travel Behavior, and Travel Carbon Emissions

The determinants and influence mechanisms of CO_2_ emissions and travel behavior are different [19]. However, research indicates that the built environment influences residents’ travel decisions, which, in turn, influences travel carbon emissions. This means that carbon emissions, as a derived outcome, and travel behavior act as mediating variables between many independent variables and carbon emissions [13,29,30]. Therefore, it is likely that the built environment variables that impact individual travel behavior will also impact CO_2_ emissions [19]. Most academics focus on travel mode selection, travel distance, travel frequency, travel purpose, and travel time when studying the relationship between the built environment and travel behavior [17,31,32,33,34]. Although the findings of these studies are not entirely consistent, they have some similarities. According to a review by Ewing and Cervero [33], the built environment has the greatest influence on how far people travel, followed by socio-demographic characteristics. The built environment and socio-demographic characteristics have the greatest influence on mode choice; however, the influence of socio-demographic characteristics may be greater. Individual socio-demographic characteristics are more likely to determine how often people travel than the built environment. The built environment has a greater impact on the total travel distance or travel time than individual socio-demographic characteristics. In a follow-up study over a ten-year period, the author [17] found that the built environment has a massive impact on travel behavior. 

In the theoretical field, behavior-centered demand analysis has become a hot issue in the field of transportation in recent years. According to the classical theory of transportation behavior, travel is a derived demand [35]. People must travel to their destinations in order to participate in activities that are in different locations. Since the built environment determines the spatial distribution of different activities and the time constraint to travel to and from different locations, it directly affects people’s activity participation and, thus, their traffic travel behavior. So far, travel behavior research has built up a framework of “disaggregate analysis—mechanism study—causal path inference” based on derived demand theory. However, the outcomes of travel behavior, such as carbon emissions, have received little attention in these studies [13,36]. The established literature suggests that the focus of existing studies on carbon emissions as an outcome of travel behavior is inadequate, that travel behavior is only a mediating variable affecting travel carbon emissions [12,13,36], and that direct research on carbon emissions is needed to obtain a threshold for the impact of travel carbon emissions. Therefore, travel carbon emissions studies are warranted [19,30]. In terms of empirical results, the determinants that influence travel behavior and travel carbon emissions also differ. In Ding et al.’s study [22] of travel distance in Oslo, location and density were the two major determinants, while in Wu et al.’s study [19] of carbon emissions from travel in Minneapolis, land use diversity and age were determinants, but both concluded that the built environment was the more important element. There are differences in the specific influence mechanisms of each element on distance and carbon emissions. Personal social attributes seem to be a more important dimension in the travel mode choice [37,38]. The specific impact mechanisms of travel mode choice are the opposite of carbon emissions. These comparisons show that, although the theoretical framework of travel carbon emission studies is derived from travel behavior studies, the determinants of travel carbon emissions and the specific impact mechanisms are different from travel behavior. These differences may be attributed to the fact that travel carbon emissions are superimposed on travel mode choice and travel distance, and this superimposed effect brings about differences in impact mechanisms. A direct reference to the findings of travel behavior studies to guide carbon reduction may be fallacious.

At present, most of the studies on travel CO_2_ emissions are based on the framework of travel behavior studies conducted in developed countries, such as North America, Europe, and Oceania. According to the research methodology, this literature can generally be divided into aggregate and disaggregate research.

Typically, aggregate studies analyze the relationship between the built environment and CO_2_ emissions at an aggregate level, using the traffic zone (or the community, street, city, or country) as the basic unit. Newman [39] compared 32 cities worldwide and concluded that residential and job density are negatively correlated with energy use. Hankey and Marshal [29] examined aggregate data from 142 cities in the USA and predicted six urban-expansion scenarios. According to the study, compact development can reduce cumulative emissions by 15–20%. Gallivan [40] examined data from more than 300 urbanized areas in 9 metropolitan areas in the USA and concluded that changing density and accessibility can reduce travel distance, energy consumption, and carbon emissions by 8%. Yang et al. [25] studied the impact of socio-demographic factors, urban morphology, and traffic development on traffic CO_2_ emissions in China from 2000 to 2012 and concluded that population density had a significantly positive effect on traffic CO_2_ emissions in China, while the level of public transportation development had a significantly negative effect on traffic CO_2_ emissions. These studies have not yet reached consistent conclusions. However, they provide strong evidence for carbon reduction policies. Because of the aggregated nature of the built environment and CO_2_ emissions data, aggregate studies can only assemble individual travel factors to a specific area, which masks differences between individuals in the same area and leads to a partial generalization of research findings [18].

A disaggregate analysis is required to address these issues. Disaggregate analysis, as opposed to aggregate analysis, where individual characteristics are lumped together in a specific area, typically uses built environment variables that focus on individuals, such as measuring the built environment within 1 km of an individual’s home address [41]. Individual travel behavior is typically expressed as a function of the built environment and socio-demographic attributes in disaggregate analysis [18]. This can provide an empirical basis for inferring the mechanism of the impact of the built environment on CO_2_ emissions and judging the relative importance of the impact of the built environment and socio-demographic attributes; thus, it has gradually become the mainstream of built environment and CO_2_ emissions research. According to the impact mechanism, the built environment influences travel CO_2_ emissions through intermediate variables such as car ownership [12,13], travel distance [12], and choice of travel mode [42]. Regarding travel behavior, most of the literature examines the impact of “5D” built environment elements (density, diversity, design, destination accessibility, and transit proximity) [43,44] on CO_2_ emissions. Some studies [23] have considered elements of transport demand management, such as bus subsidies and parking supply and charges. According to the literature, built environment factors, such as residential and job density [45,46], land use mix [47], intersection density [47], and public transportation accessibility [46], are all negatively correlated with energy consumption and carbon emissions. Research from Zahabi et al. [48] in Montreal, Canada, showed that a 10% increase in residential density, land-use mix, and transport accessibility leads to 3.5%, 2.5%, and 5.8% reductions in GHG emissions from household transport, respectively. Hong and Goodchild [49] arrived at a similar conclusion in their study conducted in Puget, USA. Transport emissions can be reduced by 31.2–34.4% by doubling the land use mix and intersection density. Hong [50] found a nonlinear relationship between residential density and travel carbon emissions, with increasing residential density having a marginal decreasing effect on carbon emissions reduction. Wu et al. [19] studied the Twin Cities of Minneapolis and St. Paul, USA, confirmed the prevailing nonlinear relationship between “5D” built environment elements and CO_2_ emissions, and examined the threshold effects of each variable on the impact of travel CO_2_ emissions.

Moreover, Chinese travel preferences are different from those in the West [51]; for example, Chinese residents prefer to commute short distances and commute by public transport, and the Chinese use of cars depends heavily on them as a sign of social status. The built environment and transport carbon emissions in China are different from those in the developed West [25,26]. On the one hand, the density of China’s large cities is generally higher than that of cities in Western countries. On the other hand, unlike Western countries that currently advocate for compact development, Chinese cities are undergoing a process of suburbanization and decentralization, and the supply of urban public transportation is unequally distributed, forcing people to choose motor vehicles for long-distance commuting in suburban areas with a single function and insufficient public transportation facilities. This dual variability in subjective travel preferences and the objective built environment may result in differences in inherent mechanisms. Academic studies of Chinese cities have come to different conclusions from those in Western countries, and several density studies conducted in China have shown that the relationship between density and CO_2_ emissions is not significant [25,52]. Ma et al. [42] and Xiao et al. [53] arrived at opposing conclusions regarding the provision of public transport. The former shows a negative relationship between public transport accessibility and CO_2_ emissions, whereas the latter has a positive impact on CO_2_ emissions. Therefore, in the context of the goal of achieving carbon peaking by 2030 and carbon neutrality by 2060, there is a great need for direct travel carbon emissions research in China. 

### 2.2. Synergistic Effect of Multidimensional Elements

As discussed in the previous section, although single variables such as the built environment, individual social characteristics, and TDM can all influence travel behavior and travel carbon emissions by themselves, integrating them may have synergistic effects [18,20]. The synergistic effect of variables has two implications for planning policy development. 

First, there is a joint effect of independent variables on dependent variables among different dimensions, such as the synergistic effect of TDM and built environment variables; TDM amplified the impact of the built environment on travel mode choice [23]. Guo et al. [54] studied the synergistic effects of Portland land use variables and congestion pricing on VMT, and they found that congestion pricing tended to have a greater impact on VMT reduction in downtown neighborhoods than in suburban neighborhoods. This is attributed to the fact that urban centers with a mixed density and mixed uses have more travel options, and when residents are charged by congestion, they can choose to reach closer destinations to avoid the congestion charge. Lee et al. [55] showed that the effect of gasoline prices on public transportation ridership was greater in areas with higher densities and more complete transit facilities. Overall, these studies of synergies between transport and land policies point to the importance of synergies between different dimensional variables over the impact of single built environment dimension variables. This means that the research horizon should not be limited to exploring the impact of the built environment but should extend to multiple dimensions. 

Second, there are synergistic effects between variables of the same dimension, such as the synergy of different individual socio-demographic variables. Research shows that women in the low-income group are more likely to choose public transport for commutes [38]; this implies a synergistic relationship between income level and gender. Yang et al. [56], in Shenzhen, concluded that land use elements and metro station accessibility have a synergistic effect on urban vitality and that increased land use intensity amplifies the positive effect of metro station accessibility on urban vitality. This can provide empirical evidence for urban TOD development. In addition, synergistic effects are important for the capture of specific thresholds. For example, in the case of land use policy, we derived a threshold for land use diversity, but this single threshold may not be accurate due to the joint effects and complex interactions between different independent variables. For land use, the two aspects of land use intensity and land use mixture are the most important. We prefer to know the optimal combination of the values of these two key elements, and capturing the synergistic effects of variables helps us to address this issue. 

### 2.3. Research Gaps

Although established studies have explored the nonlinear effects of the built environment on travel behavior, such as travel mode choice, travel distance, and travel frequency, travel behavior has been shown to be only a mediating variable of carbon emissions [3,12,13,57], and travel carbon emissions are a derivative of the combined effects of multiple travel behaviors. This causes the findings of carbon emission and travel behavior studies to differ in terms of the relative importance of variable effects, impact effects, and threshold relationships. However, there is a limited amount of literature addressing commuting carbon emissions. Recently, a few studies have attempted to investigate the complex mechanism of the built environment’s influence on CO_2_ emissions in Chinese cities using methods such as structural equation modeling and geographic weighting regression [3,12,13,58]. However, these methods rely on linear assumptions, and few studies have focused on the nonlinear relationship between them. Indeed, accurately identifying this nonlinear relationship is critical because the threshold effect implies that land-use policies for reducing carbon emissions by changing built environment variables may only be effective within a range of values [22,23], and the cost of policy implementation can be significantly reduced by identifying a reasonable range and threshold [59].

Alternatively, most existing studies have considered the impact of a single built environment variable on carbon emissions, while few have considered the synergistic effect of multiple variables. As the growth rate of China’s urban expansion decelerates, planners formulating land use and transport policies need to determine a reasonable range of indicators for other elements of the built environment, given that some elements (e.g., location, population density, land use intensity, and socio-demographic characteristics of the population) are established. Identifying the differential impacts of the built environment and transportation policy characteristics under different planning scenarios can provide more subtle implications for planning and policy development [23,56]. Traffic congestion and emissions are becoming increasingly serious as China’s motorization continues. Parking charges, transport subsidies, congestion charges, and other transport demand management policies have been widely used in urban transport management. However, the interactions between these policies and traditional land use policies in terms of carbon emissions are poorly understood. Furthermore, with little consideration of the urban physical geography grid, all previous studies have assumed that built environment factors are distributed over an idealized homogeneous surface. However, these factors must not be overlooked. For example, cross-river commuting may influence the mode of transportation of residents, thereby influencing CO_2_ emissions. How do these characteristics relate to commuting carbon emissions? Do they have a synergistic effect on commuting carbon emissions when combined with other “5D” built environments?

To address these issues, this study uses a large global sample dataset in Wuhan, China, and a machine learning method to investigate the nonlinear and synergistic effects of built-up environmental factors on commuting carbon emissions in a Chinese metropolis.

## 3. Methodology

### 3.1. Research Area and Data Source

Wuhan was chosen as a case study for two reasons. First, Wuhan is representative; it is the largest metropolis in central China and has the typical high-density and outward expansion characteristics of Chinese cities. In the past two decades, Wuhan has experienced rapid urbanization and motorization. The population density of the Wuhan metropolitan area has reached 5898–25,790 people/km^2^ (regional average), and the built-up area of Wuhan grew by 365 km^2^ from 2012 to 2022(Figure 1). There was intense pressure to reduce transport emissions, as Wuhan’s transport carbon emissions increased at an average annual rate of 11.6% from 2005 to 2017. Secondly, Wuhan is unique in that it is divided by two major rivers, the Yangtze River and the Han River, and its unique geographical features create a large demand for cross-river commuting, which may have a significant impact on commuting carbon emissions [60]. Wuhan can therefore serve as an excellent reference for other cities, especially those of a similar urban size, urban form, and traffic structure.

The travel data in this study came from the fourth travel survey of residents in 2020 conducted by the Wuhan Institute of Transportation Development Strategy. Surveys have been conducted every decade to determine the basic characteristics of the daily movements of the population. A structured Family Interview Questionnaire was used to collect daily travel information from residents, such as the start and end points of daily travel, the purpose of travel, the selected mode of travel, and the social and economic characteristics of families and individuals, including age, gender, personal education level, employment status, household registration, family income, family size, and number of children. The survey was conducted in Wuhan, with a 0.5% sample rate (Table 1). During the two-month period from October to December 2020, surveyors randomly selected a sample of every 10 households for face-to-face household interviews according to the principle of equidistant sampling and used WeChat applets to fill in the questionnaire. Researchers conducted random household surveys using the WeChat app. A total of 43,660 people’s travel information was collected over a two-month period. The data were further processed, missing information was removed, and 29,291 samples were obtained.

The preliminary statistics of the survey results show that, in terms of the structure of motorization modes, the share of public transportation in Wuhan in 2020 was 50.3%, and the share of car trips was 45.1%. Compared with 2008, when public transportation in the main urban area accounted for 54.5%, car trips accounted for 25.9%, and other motorized trips such as cabs and motorcycles accounted for 20.0%, the share of public transportation decreased slightly, while the share of car trips and trips increased greatly. The number of private cars increased from 308,000 to 3.159 million from 2008 to 2020. The number of private cars increased by about nine times, which is the main reason for the significant growth of small cars and other modes of transportation. In terms of travel purposes, commuting trips account for the largest share: about 37.7%. The city’s average travel distance also increased from 5.1 km in 2008 to 6.0 km. The city’s average commuting travel distance reached 8.5 km in 2020, and the average motorized travel distance was 10.5 km. 

### 3.2. Variables and Data

The origin and destination (OD) points of residents’ commuting trips obtained from the residents’ travel survey were vectorized, and the Baidu Map Path Planning API was used to calculate the actual commuting distance of a single sample. The algorithm calculates distances based on the respondents’ OD point addresses and according to the principle of optimal route decision (considering the shortest travel time, the shortest travel distance, the travel comfort, and the travel cost), which is the closest to the respondents’ real commuting scenarios. Its commuting carbon emissions are based on the individual commuting modes obtained from the survey. For the calculation, the following formulae were used:(1)$$C_{i}=\sum C_{\mathrm{ij}}$$
(2)$$C_{\mathrm{ij}}={\;\mathrm{MTD}}_{\mathrm{ij}}\times {\;\mathrm{EF}}_{m}$$
(3)$${\mathrm{MTD}}_{\mathrm{ij}\;}={\;\mathrm{TD}\;}_{\mathrm{ij}}-{\;\mathrm{NTD}}_{\mathrm{ij}}$$

C_i_ is the daily carbon emissions from individual commuting, which is the total amount of carbon emissions generated by individual commuting in a day. C_ij_ is the carbon emissions from commuting trip j for sample i, and EF_m_ is the carbon emission coefficient of residents’ commuting vehicles. This study refers to the Carbon Emission Intensity Table for Mass Transportation in China. Xiao et al. [53] calculated the carbon emissions of the daily travel of Beijing citizens and compiled the carbon emission factors applicable to this study (Table 2). To estimate the carbon emissions of the commute as accurately as possible, it is necessary to compute the pure motorized distance of this regular sample i trip. Consequently, when an M-class regular trip is used to complete trip j, the “Baidu Map Batch Road Calculation API” is used to calculate the travel distance (TD_ij_) of the shortest travel scheme. This section of travel distance includes both motorized and non-motorized travel distances. NTD_ij_ represents the non-motorized travel distance in this section, and MTD_ij_ represents the pure motorized travel distance.

In addition, using the Baidu Map API and the actual path, the actual range of the residents’ 15 min walk (15 min isochronous circle) was calculated as the exact range of the built environment index around the statistical residence. Existing studies primarily use communities, streets, traffic districts [3,12,13], or buffer zones at a certain distance from the residence [19,41] as boundaries to measure the built environment around individuals; however, these boundaries cannot accurately reflect residents’ actual daily activities. Errors may occur in the measurement of the built environment around an individual when using these simplified “data boundaries”. This leads us to replace these common boundaries with the isochronous circle of actual travel.

The Wuhan Urban GIS database provides measured data for built environment variables. Among them, dwelling location characteristics were expressed by the Euclidean distance between the dwelling and the city public center and the nearest cluster center, and the cluster center was identified by Wuhan mobile phone LBS data [61]. The mixed entropy index of land use considers six land types: residential, commercial, educational, industrial, public services, and green space. Data on the resident and employed populations were obtained from the Wuhan census. Furthermore, the residents’ transportation subsidies and parking facility use were obtained from the questionnaire.

The statistical table of the variables involved in the study is as follows (Table 3).

### 3.3. Modeling Methods

The random forest (RF) model was used in this study to uncover the complex relationship between multidimensional elements and commuting carbon emissions. Ho [62] proposed RF, which is based on the integration method of decision trees and optimizes model fitting and prediction by assembling a large number of individual decision trees [63]. It extracts multiple samples from the original samples using the bootstrap resampling method and builds a decision tree model for each bootstrap sample. The split in each tree lasts until the tree reaches its maximum depth. The forecasts of multiple decision trees are then combined, the final forecast is obtained through voting, and the final result is obtained by averaging the forecasts of all individual trees. The working route is shown in Figure 2.

To enhance the interpretability of the model, the random forest model can generate partial dependence plots (PDP) to illustrate the relationship between a single independent variable and the dependent variable while considering its interactions with other independent variables. This function can be used to assess the nonlinear relationships between variables through multivariate analysis. In addition, it can be used to assess the interaction effects between two or more independent variables, which is another important model function used in this study.

Numerous theoretical and empirical studies have shown that RF is highly predictive, robust regarding outliers and noise, and not prone to overfitting [64,65]. Most importantly, it directly expresses the true relationship between variables rather than assuming a specific parameter relationship between the self and dependent variables, as in traditional linear regression [66]. The RF algorithm can handle missing values and maintain accuracy if some data are lost. In addition, the RF algorithm performs better on a large-sample dataset and is less sensitive to outliers than another popular machine learning algorithm, GBDT. The random forest model generalizes better when dealing with datasets with larger samples (GBDT is prone to overfitting), and the estimation of parameters takes less time. (Due to the sample size being close to 30,000, the parameter optimization for GBDT is unacceptably time-consuming.)

For model calibration, three parameters must be considered: the total number of trees n (forest size), the number of split variables m, and the maximum tree depth d [67]. Alternatively, it can calculate the relative importance of a single variable in a variable dataset and create partial dependency plots to show the relationship between the independent and dependent variables. Furthermore, in recent years, some researchers have questioned whether the higher fit of RF compared to linear forest is due to overfitting. This study used a dataset with a larger sample size (29,291) to avoid overfitting.

## 4. Results

This study used Python3.8 to build an RF model, with 20% of the samples randomly selected as the test set and 80% as the training set, in order to obtain more reliable model results. The parameters were optimized in two steps using Bayesian estimation with the Hyperopt algorithm: (1) determine the best decision tree scale n, and (2) determine the best split variable number m and tree depth d. Finally, after 300 iterative optimizations, optimal model parameters are determined. The results show that the model’s predictive performance can be neglected, and more time is consumed after more than 241 trees; therefore, the number n of trees was set to 241. The model’s prediction performance is best when the maximum tree depth d is 35 and the split variable m is 5. Furthermore, the RF model was found to have a better goodness of fit and generalization when comparing the regression results of the GBDT model. The final model’s R2 was 0.51 (the R2 of the linear regression model with the same independent variable dataset was 0.29 (Appendix B), and the GBDT’s R^2^ was 0.40 (Appendix A)), which was used to quantify the importance of each built environment variable and draw partial dependency plots.

### 4.1. The Relative Importance of Independent Variables

Table 4 depicts the relative importance of the built environment characteristics, socio-demographic characteristics, and traffic demand management policies. The total relative importance of all the predicted variables was 100%. In general, residents’ socio-demographic characteristics accounted for only 18.75% of the total, while the built environment (60.21%) and traffic demand management indicators (21.04%) played an important role in predicting commuting carbon emissions. Notably, the availability of free parking in the workplace contributes 18.29% of the importance, indicating that it has a significant impact on commuting carbon emissions. The contribution of “whether or not to commute across rivers” ranks second at 11.09%, indicating that, in Wuhan, a megalopolis with riverine development, cross-river commuting behavior has a significant impact on commuting carbon emissions, and the inconvenience of non-motorized commuting caused by river barriers increases the possibility of motorized travel.

In addition, job and residential density contributed the most, at 6.32% and 5.41%, respectively. Location is also important, with the distance to the city center and the distance to the nearest cluster center contributing 6.15% and 5.10%, respectively. This is similar to the conclusion of Ewing and Cervero’s review [17]: people living in higher-density areas tend to have shorter commutes, as a higher density can encourage non-motorized travel and the use of public transport. Location largely determines the characteristics of the built environment around a place of residence [68]. For example, people living near urban centers have shorter commuting distances and are more likely to choose to commute by public transport because urban centers have a higher concentration of employment and transport services. In this respect, we differ from Zhang et al. [69], who studied the impact of multi-centers on travel mode choices in Beijing. For Wuhan, the distance from the city center affects carbon emissions more than the distance from the cluster center. This shows that, although Wuhan has recently embraced multi-center planning to reduce commuting demand and distance, the attraction of the original powerful center remains considerable. Moreover, the impact of land-use diversity in Wuhan was less significant (4.44%, ranked 11th), in contrast to the findings of Wu et al. [19], who studied carbon emissions from travel in the Twin Cities of Minneapolis using a similar methodology (GBDT). This could be due to the fact that Wuhan has a higher level of land use diversity (0.68 on average) than Minneapolis (0.53 on average), which makes the marginal impact of land use diversity on carbon emissions less obvious. Therefore, reducing CO_2_ emissions by adjusting land-use diversity in Wuhan may be ineffective.

According to a review by Ewing and Cervero [33], the built environment is more important for travel distance, followed by socio-demographics, and mode choice depends on both. It is possible that the built environment has a greater impact on travel CO_2_ as a function of both. In addition, a more recent study by Ewing [17] found that the comprehensive effect of built environment variables on travel behavior was substantial. Furthermore, several studies have found the built environment to be more relevant to travel behavior using nonlinear models. They generally agree that nonlinear models (such as decision trees, random forests, GBDT, and GAM) can accurately capture the potential threshold effects of built environment variables on carbon emissions, whereas ignoring these subtle effects can lead to an underestimation of the impact of the built environment. Second, nonlinear models have the advantage of capturing as much of the influence of built environment variables as possible because they can automatically capture interactions between variables and help deal with multicollinearity. Thus, it is reasonable to conclude that the built environment has a greater impact. The study also found that parking fees, transit subsidies, and other transportation demand management (TDM) policies cannot be ignored on this basis. We must examine the specific impact of these variables on commuting carbon emissions.

### 4.2. Nonlinear Influence of a Single Variable on Commuting CO_2_ Emissions

Using an RF approach, partial dependence plots were developed to explain the association between commuting carbon emissions and the built environment, individual characteristics, and TDM.

Figure 3 depicts the effect of density on commuting CO_2_ emissions. As shown in the partial dependence plots, both residential and job density have a negative effect on commuting carbon emissions. The threshold for residential density is approximately 23,000 people/km^2^. Commuting carbon emissions reached their lowest point at a residential population density of 23,000 persons/km^2^, at which point the trend of carbon emission reduction slowed until a weak positive correlation was observed. Commuting carbon emissions are the lowest at a job density of approximately 5000 persons/km^2^, and then marginal effects become weak. This demonstrates that, while density is negatively correlated with commuting carbon emissions, this negative correlation does not persist and disappears or becomes weakly positive above a certain threshold. Notably, the density threshold obtained in this study was markedly higher than that obtained by Wu et al. [19] in Minneapolis, owing to the high density in Wuhan. The population density of the Wuhan metropolitan area is 5898–25,790 people/km^2^ (regional average) [70], and the population density of local communities exceeds 60,000 people/km^2^, which is significantly higher than that of Minneapolis (average 2750 people/km^2^). This demonstrates the importance of conducting relevant research in developing countries such as China, because the experience of Western developed countries may not apply to China. Currently, the population density in Wuhan’s city center is above the threshold obtained in this study. Therefore, for densely populated urban areas, increasing population density is unsuitable for reducing CO_2_ emissions from transport. The population of urban centers should be controlled to avoid additional commuting CO_2_ emissions due to the excessive population concentration in urban centers, while the functions and population should be relieved to cluster centers with low population densities.

Figure 4a depicts the effect of the distance to the city center on CO_2_ emissions from commuting. The effect of the distance to the nearest cluster center on commuting CO_2_ emissions is shown in Figure 4b. Using Wuhan mobile LBS data, five cluster centers were identified in Wuhan, and these urban clusters are also key development targets identified in Wuhan’s recent polycentric planning. Overall, the distance to the city center has a positive relationship with carbon emissions from commuting. This is because the farther the place of residence from the city center, the longer the commuting distance of the residents. Consequently, people prefer to drive long distances to save time compared to public transport. The closer to the center, the shorter the commute, and the higher the level of transit is, the more green transportation options (e.g., walking, bicycling, and public transit) there are, reducing commuting CO_2_. For centers at different levels, commuting carbon emissions begin to increase sharply beyond 8 km from the old city center, and the marginal impact disappears beyond 16 km. However, commuting carbon emissions increase when the distance to an adjacent cluster center exceeds 4 km. This phenomenon illustrates two issues. First, the low-carbon commuting sphere of influence of the top urban center is around 8 km, beyond which the attractiveness of green transportation modes decreases, leading to an increase in commuting carbon emissions until the distance reaches 16 km, when the top urban center is replaced by other low-level centers such that the marginal impact disappears or even becomes negative. Second, cluster centers have a sphere of influence of approximately 4 km, which is smaller than that of urban centers. It has a certain substitution effect on urban centers; however, in terms of partial dependence and contribution, urban centers clearly have a greater influence on commuting carbon emissions. This indicates that, despite the recent trend of Wuhan developing in a multi-center mode, the occupation and residence functions of each cluster center have not yet been fully developed and cannot replace the top-level urban center.

Figure 5 depicts the nonlinear effects of other built environment factors on commuting carbon emissions, including land use diversity (Figure 5a), intensity (Figure 5b), road density (Figure 5c), intersection density (Figure 5d), and bus accessibility (Figure 5e,f). These variables have nonlinear effects on carbon emissions from commuting. The impacts of land-use intensity and diversity on commuting CO_2_ emissions are depicted in Figure 5a,b. Most existing studies suggest that land-use mix and development intensity are negatively correlated with travel CO_2_ emissions because more intensive and mixed development encourages public transit use and non-motorized travel [17]. However, our conclusions are different. The partial dependence plot of land development intensity and land use mixing degree demonstrates that this negative correlation exists but only at the threshold. The land development intensity threshold was approximately 3.5, and the land-use mixing degree threshold was approximately 0.68. Personal commuting CO_2_ can be reduced by 80 g per day as the mixing degree increases from 0.5 to 0.68; however, when it exceeds the threshold, the negative correlation disappears. This demonstrates that, for a low-carbon land use policy, the degree of land use mixing and intensity should not be increased indefinitely, as excessive density and mixing may lead to increased commuting demand, offsetting the original negative effects. Furthermore, the density of the road network and the density of intersections are negatively correlated with commuting CO_2_ emissions over a wide range, and denser road networks and intersections can increase road connectivity and promote non-motorized public transportation. However, the road intersection density threshold was 15 per square kilometer, and the road network density threshold was 8 km/km^2^. Beyond these values, the negative correlation weakens and even turns into a rapid positive correlation.

For public transit (Figure 5e,f), when the number of stops within a resident’s 15 min isochronous circle exceeds 37, the negative correlation between the number of stops and commuting carbon emissions changes to a sharp positive correlation (Figure 5e), indicating that more bus stops are not better for carbon reduction. The same pattern is observed in the effect of distance to the nearest bus stop on carbon emissions, where distance is negatively or weakly correlated with carbon emissions in the linear distance range of 0–180 m (Figure 5f). Above 180 m, the carbon emissions begin to increase sharply. This indicates that when the linear distance to the stop exceeds 180 m, public transit becomes significantly less attractive to commuters because they have to walk or ride a distance to reach the transit stop (this actual distance may be much greater than the linear distance of 180 m), which is not convenient enough for commuters; therefore, commuters will prefer to commute by car if the nearby public transit stop is too far from where they live.

Figure 6 depicts the impact of age and traffic demand management policies as well as cross-river commuting on carbon emissions from transportation. The age of commuting CO_2_ first has a negative correlation and then has a positive correlation after 25 years old (Figure 6a). Providing bus, fuel, and taxi subsidies increased commuting carbon emissions; however, the positive correlation of bus subsidies was weak. Subsidies for gasoline or taxis increased carbon emissions from 750 g per person per day to approximately 1400 g per person per day (Figure 6b). Similarly, cross-river commuting and free parking both significantly increase commuting carbon emissions (Figure 6c,d).

### 4.3. Synergistic Effects of Different Variables on CO_2_ Emissions

The synergistic effect of cross-river commuting behavior and other built environment variables on commuting CO_2_ emissions is shown in Figure 7. First, the partial dependence plots show that cross-river commuting has a significant positive effect on commuting carbon emissions. For example, with the same population density of 19,116 persons/km^2^, the partial dependence plot shows that residents with cross-river commuting emitted 1390,31 g of extra CO_2_ compared to those without cross-river commuting (Figure 7a). Furthermore, cross-river commuting and residence location had a clear synergistic effect on carbon emissions from commuting. A similar phenomenon exists when synergizing with other built environment variables (Figure 7b,c). Without cross-river commuting, the distance to the city center increased from 3.77 km to 15.84 km, individual commuting carbon emissions increased by 169.52 g per day, and with cross-river commuting, the increase in individual commuting carbon emissions was 846.11 g (Figure 7d). In other words, cross-river commuting amplifies the impact of the residential area location. This is because Wuhan is divided into three cities, Hankou, Hanyang, and Wuchang, by the Yangtze and Han rivers, which are connected by a limited number of river bridges and do not have convenient pedestrian and bicycle lanes; therefore, even if the commuting distance is short, it is inconvenient and unsafe to use non-motorized means to cross the river. Therefore, commuters prefer to drive across the river to reach the city center, which increases commuting CO_2_ emissions.

Figure 8 depicts the additive effects of free parking and other environmental factors on commuter CO_2_ emissions. When free parking is provided at the destination, there is a substantial increase in individual commuting carbon emissions (Figure 8a–f), suggesting that free parking facilities increase the likelihood of motorized travel, which in turn significantly increases commuting carbon emissions. Second, when free parking is not provided, the effects of some built environment variables are reduced. The synergetic partial dependence plot of the number of bus stops and free parking, for example, shows that as the number of bus stops increases from 0 to 54, the carbon emission from commuting is only reduced by 37.90 g without free parking; however, it can be reduced by 247.14 g with free parking (Figure 8f). It is demonstrated that free parking and other built environments have a synergistic effect on commuting carbon emissions. In other words, the parking fee policy can partially compensate for the effects of changes in the built environment. Setting a parking fee policy is an effective and low-cost method for reducing individual commuter carbon emissions. In the absence of a parking fee policy, the second-best means of reducing commuting carbon emissions by changing the built environment has been effective to some extent.

Figure 9 depicts two important land-use indicators: the synergistic effect of the land-use mixing degree and land development intensity on commuter CO_2_ emissions. The predicted findings show that individual carbon emissions for commuters are the lowest at 709.73 g when the land mix entropy index is 0.70 and the land use intensity is 4.00 (Figure 9a). Furthermore, the circle of lowest commuting CO_2_ emissions forms when the land mix entropy indexes are within the range of 0.66–0.73, while the land use intensities are within the range of 3.12–4.41, indicating that a moderate mixed land-use mode combined with appropriate high-intensity land development is beneficial for reducing CO_2_ emissions. This provided a specific numerical basis for policy development. In addition, a bivariate partial dependence plot based on road network density and land development intensity shows that a reasonable neighborhood scale around the residence and an appropriate volume ratio can significantly reduce commuting CO_2_ emissions. The combination of a road network density of 9.41 km/km^2^ and a floor area ratio of 4.00 results in the lowest individual commuting CO_2_ emissions (Figure 9b). The road network density of Wuhan in 2021 was 6.2 km/km^2^, indicating that the city’s road network still needs to be encrypted to reduce carbon emissions. In recent years, Wuhan has controlled the floor area ratio of the central urban area, which does not exceed 3.0 and should be slightly adjusted upward. Simultaneously, other indicators of the built environment have similar synergistic effects (Figure 10a–f).

## 5. Discussion and Policy Recommendations

In terms of theory, there are many established studies that focus on travel behavior but few studies that transfer the nonlinear premise to travel carbon emissions. Travel carbon emissions are a combined outcome of travel behaviors such as travel mode, travel distance, car ownership, and residence self-choice. Our study demonstrates the applicability of the nonlinear research framework to products derived from travel behavior. This framework can be used to conduct similar studies in the future, such as travel satisfaction and transit satisfaction studies. This study is a further extension of the origin framework of nonlinear studies on travel behavior. This study demonstrates that the study of travel behavior and its derivatives should not be limited to the built environment, and the research horizon should be further expanded.

The empirical results of this study have important implications for megacity planning in China. China plans to achieve “peak carbon” by 2030 and “carbon neutrality” by 2060. Wuhan intends to limit its carbon emissions to 173 million tons by 2022 and house 16.6 million permanent residents by 2035. If the future population and employment growth can be relatively concentrated (within 8 km of the city center and 4 km of the nearest cluster center) rather than sprawled, the CO_2_ emissions associated with individual commuting can be minimized. Currently, Wuhan plans to build 17 new cross-river passages by 2025; however, research shows that cross-river commuting caused by natural river divisions is unfavorable for reducing CO_2_ emissions, and newly built cross-river passages of motor vehicles may instead aggravate CO_2_ emissions. Therefore, the right approach should be to increase non-motorized and public transportation facilities across the river and to plan a compact and multi-centered urban spatial structure in accordance with the natural geographic pattern of the city in order to minimize the need for commuting across the river, rather than trying to “stitch” fragmented plates together with the river crossing to form a larger single center. In addition to the built environment, the role of TDM transportation demand management in carbon reduction is also clear. Parking regulation, appropriate parking charges, and the reduction of fuel and taxi subsidies can help reduce commuting carbon emissions. In addition, in recent years, Wuhan has been committed to improving public transportation and spatial quality to promote green travel for residents to reduce their travel CO_2_ emissions, and the results of our empirical study can provide precise guidance for specific planning efforts.

The findings of this study can be applied to other developing countries with urban conditions similar to those in China. This study demonstrates that the relationship between China’s built environment and commuting CO_2_ emissions differs from the conclusions reached in developed countries. However, controlling urban sprawl, developing moderately compact and mixed land around public transportation facilities, and combining it with appropriate TDM policies are all critical for lowering CO_2_ emissions.

## 6. Conclusions

This study uses a large sample disaggregate dataset of 29,291 samples covering the entire area of Wuhan and employs the RF method to investigate the nonlinear and synergistic relationships between multidimensional elements and commuting CO_2_. It specifically contributes to the literature in three ways: (1) the nonlinear relationship between the urban built environment and commuting CO_2_ emissions in China is precisely described, further challenging the linearity assumption in the established literature and comparing the results with developed countries. (2) The relative importance of multidimensional influencing factors was further assessed. (3) The synergistic effect of the built environment and traffic demand management policies on the CO_2_ emissions from commuting was investigated. In general, it fills the gaps in the research on the nonlinear correlation between the urban built environment and commuting CO_2_ emissions in China, as well as research on the synergistic effects of different variables on commuting CO_2_ emissions.

This study contributes significantly to the literature on low-carbon city planning in China. First, it demonstrates that the built environment (60.21%) has a greater overall impact than individual characteristics (18.75%) on commuting CO_2_, and the proportion of the importance of the built environment is greater than that of the research conclusions of developed countries. The conclusion that the built environment is more important is relatively reliable because more samples are used, and the samples cover urban developed and suburban areas. The impact of TDM policy on commuting carbon emissions was also found to be significant (21.04%). Free parking at the destination contributed 18.29% of importance. Because parking is usually a significant cost in large cities, if free parking spaces (formal or informal) are available at commuter destinations, commuters can disregard the cost of parking and are thus more likely to choose motorized travel, which contributes to increased commuting carbon emissions. In other words, if parking spaces are regulated and a fee is charged, commuting by car may be significantly reduced due to the cost of travel, which is a low-cost and effective way to reduce carbon emissions. It also illustrates the important role of TDM in carbon reduction in a particular dimension. Furthermore, the study details the nonlinear effects of various built environment variables on commuting CO_2_ emissions and quantifies the synergistic effects of factors such as TDM, location, and land use on commuting CO_2_ emissions. Specifically, the cut-off values for population density and job density are 23,000 persons/km^2^ and 5000 persons/km^2^, respectively; the distance to the city center has two cut-off values of 8 km and 16 km, respectively, while the distance to the nearest cluster center has a cut-off value of 4 km; for land use, the cut-off value for the mixed entropy index of land use is 0.67, while the cut-off value for land use intensity is 3.50, beyond which the marginal utility reverses; for design, the cut-off value for road network density is 9 km/km^2^, while the cut-off value for intersection density is 15/km^2^; for public transportation, the cut-off value for the number of bus stops is 38, and the cut-off value for the distance to the nearest bus station is 180 m. Commuter allowance, free parking, and cross-river commuting are all strongly associated with increases in carbon emissions, and all of these elements have synergistic effects with built environment variables.

This study, however, has several limitations: (1) because the study uses cross-sectional data from a residential travel survey, the findings of this study are closer to an “association” than to a causal “effect.” This is similar to most current studies and should be further explored in the future using data over time. (2) The measurement range of the built environment in this study was a 15 min isochronous circle in which residents walked around their residences. In the future, the impact of the built environment around residents’ workplaces on CO2 emissions, as well as the impact of the built environment on CO2 emissions at different scales, should be investigated. (3) The measurement and calculation of commuting CO2 emissions did not consider carpooling. This was due to the method of obtaining the questionnaire, and we were unable to obtain relevant information about sharing because our method was the same as that in most of the literature. (4) Limited by data, commuting distance is calculated based on residents’ commuting OD points according to the principle of optimal path (considering time, distance, comfort, and cost), and this calculation has limitations. For example, different emission factors for different types of vehicles should be taken into account, along with the impact of terrain (uphill and downhill) on carbon emissions and the impact of different energy types on carbon emissions.

## Figures and Tables

**Figure 1 ijerph-20-01616-f001:**
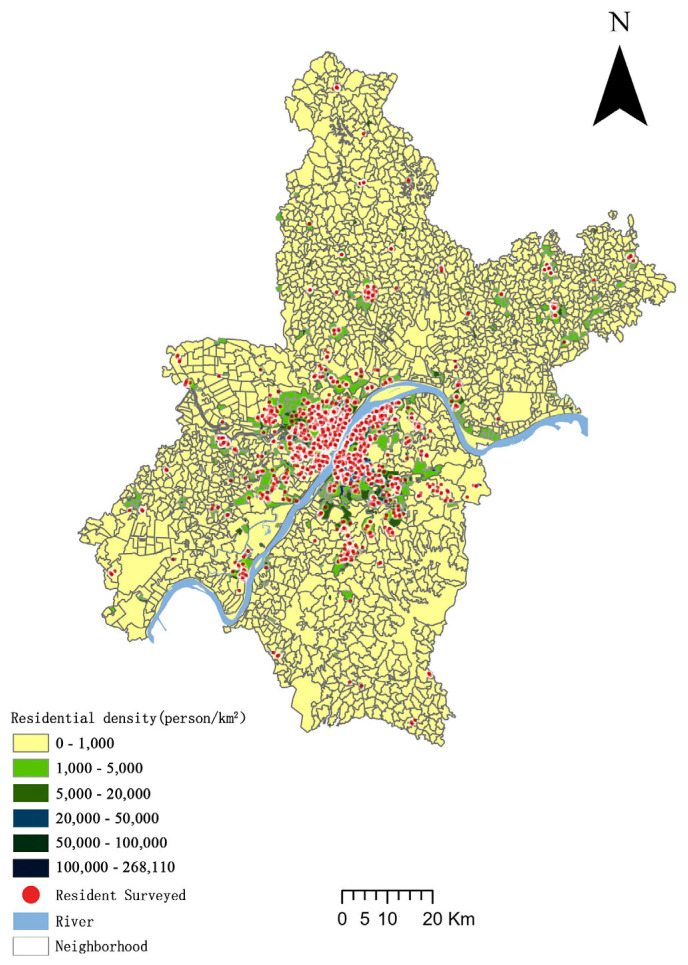
Spatial distribution of the residents surveyed.

**Figure 2 ijerph-20-01616-f002:**
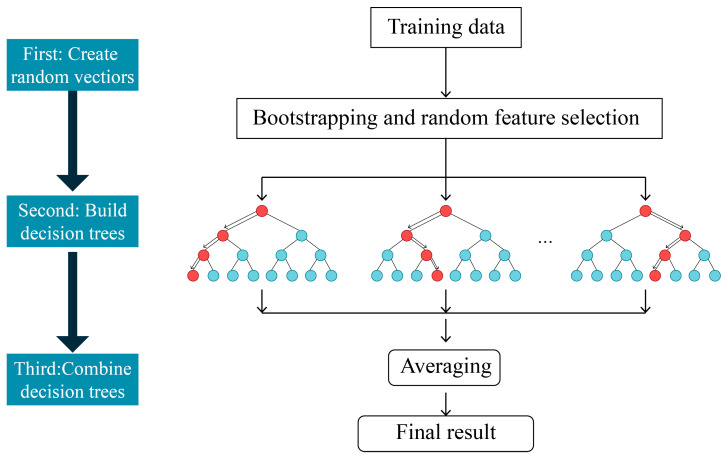
Working route of the random forest algorithm. Source: Cheng et al., 2020 [38].

**Figure 3 ijerph-20-01616-f003:**
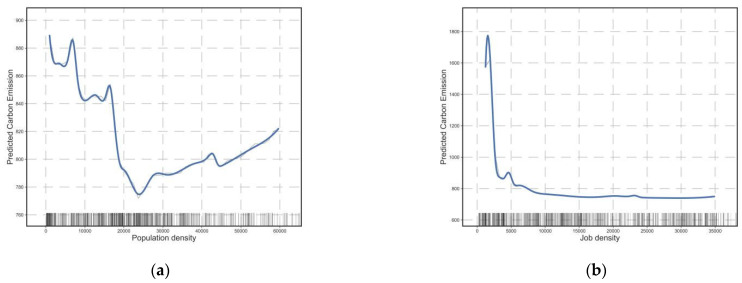
Influence of density on commuting CO_2_. (**a**) The relationship between population density and commuting CO_2_; (**b**) The relationship between job density and commuting CO_2_.

**Figure 4 ijerph-20-01616-f004:**
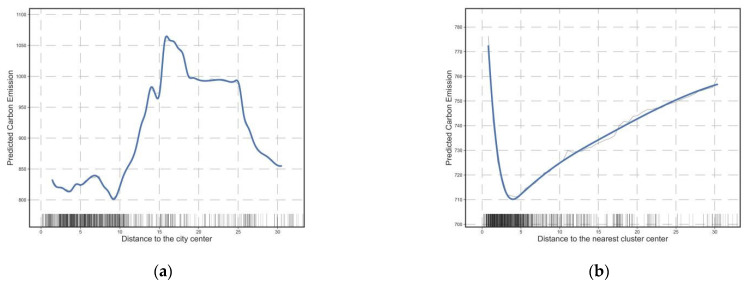
Influence of residential location on commuting CO_2_. (**a**) The relationship between the distance to the city center and commuting CO_2_; (**b**) The relationship between the distance to the nearest cluster center and commuting CO_2_.

**Figure 5 ijerph-20-01616-f005:**
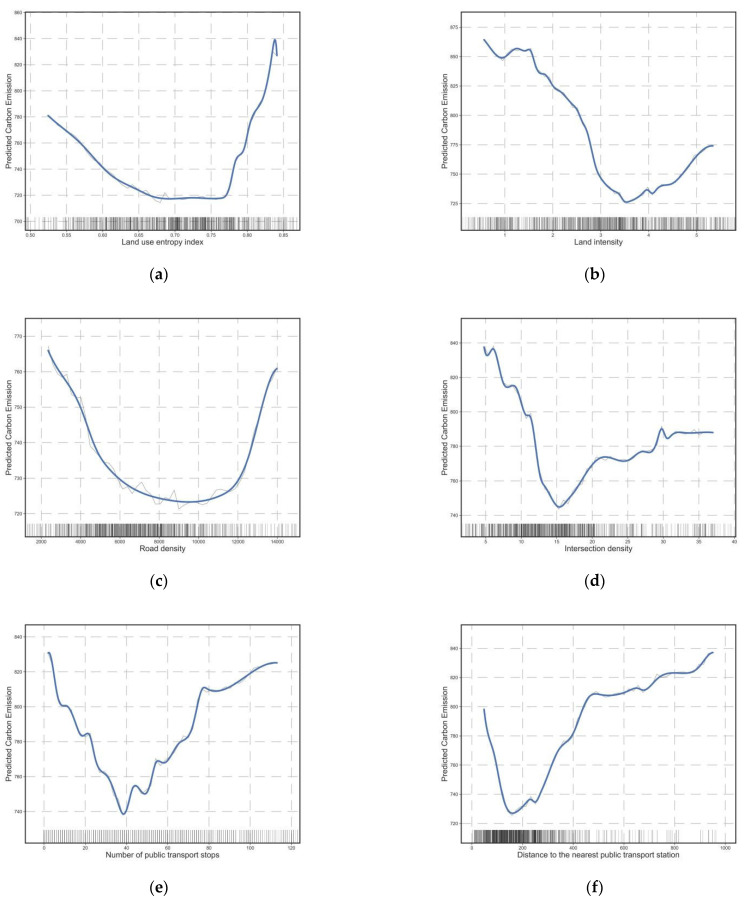
Influence of other built environment factors on commuting CO_2_. (**a**) The relationship between land use entropy index and commuting CO_2_; (**b**) The relationship between land development intensity and commuting CO_2_; (**c**) The relationship between road intensity and commuting CO_2_; (**d**) The relationship between intersection density and commuting CO_2_; (**e**) The relationship between the number of public transport stops and commuting CO_2_; (**f**) The relationship between the distance to the nearest public transport station and commuting CO_2_.

**Figure 6 ijerph-20-01616-f006:**
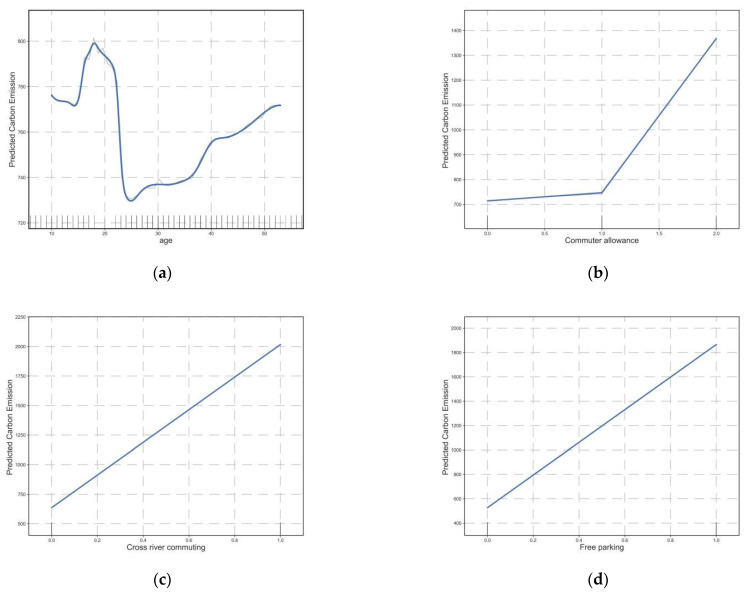
Influence of other dimensional elements on commuting CO_2_. (**a**) The relationship between age and commuting CO_2_; (**b**) The relationship between commuter allowance and commuting CO_2_; (**c**) The relationship between cross river commuting and commuting CO_2_; (**d**) The relationship between free parking and commuting CO_2_.

**Figure 7 ijerph-20-01616-f007:**
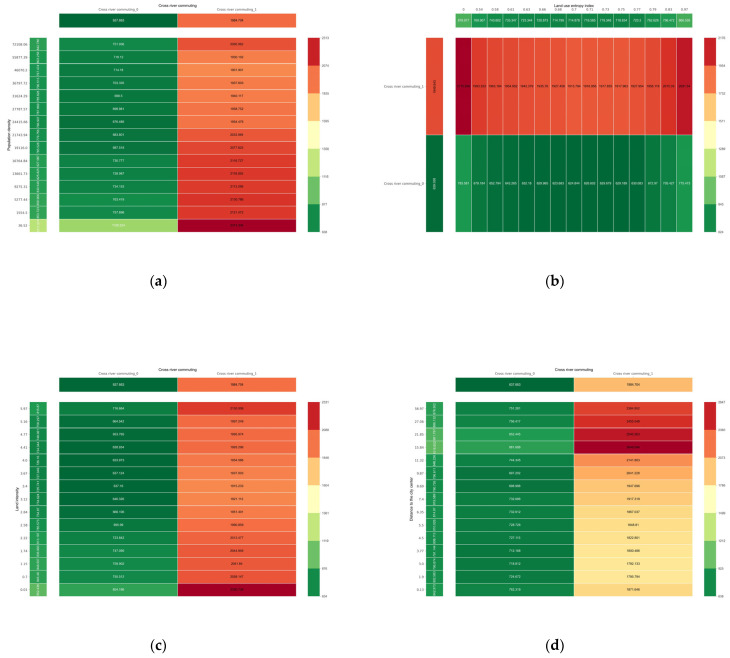
Synergistic effects of cross-river commuting with other built environment elements. (**a**) Synergy of population density and cross-river commuting; (**b**) Synergy of land use entropy and cross-river commuting; (**c**) Synergy of land use intensity and cross-river commuting; (**d**) Synergy of the distance to the city center and cross-river commuting.

**Figure 8 ijerph-20-01616-f008:**
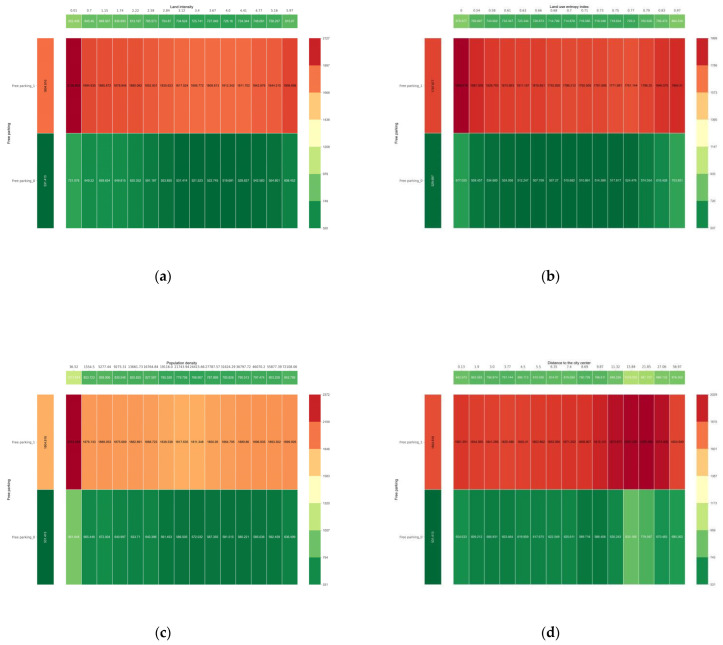
Synergistic effects of free parking and other built environment elements. (**a**) Synergy of land intensity and free parking; (**b**) Synergy of land use entropy and free parking; (**c**) Synergy of population density and free parking; (**d**) Synergy of the distance to the city center and free parking; (**e**) Synergy of road density and free parking; (**f**) Synergy of the distance to the nearest public transit station and free parking.

**Figure 9 ijerph-20-01616-f009:**
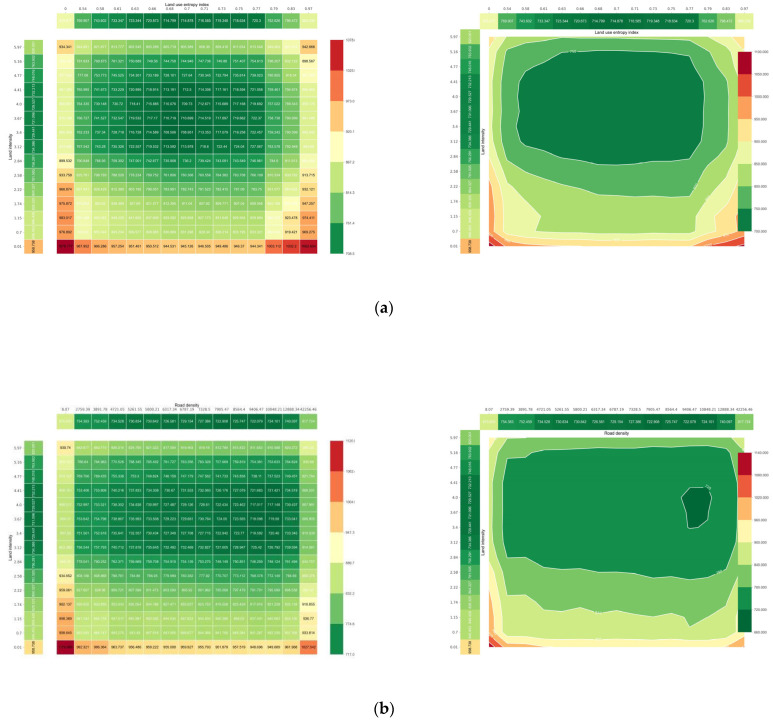
Synergistic effects of land-use indicators on commuting CO_2_ emissions. (**a**) Synergy of land use entropy and land use intensity; (**b**) Synergy of road density and land use intensity.

**Figure 10 ijerph-20-01616-f010:**
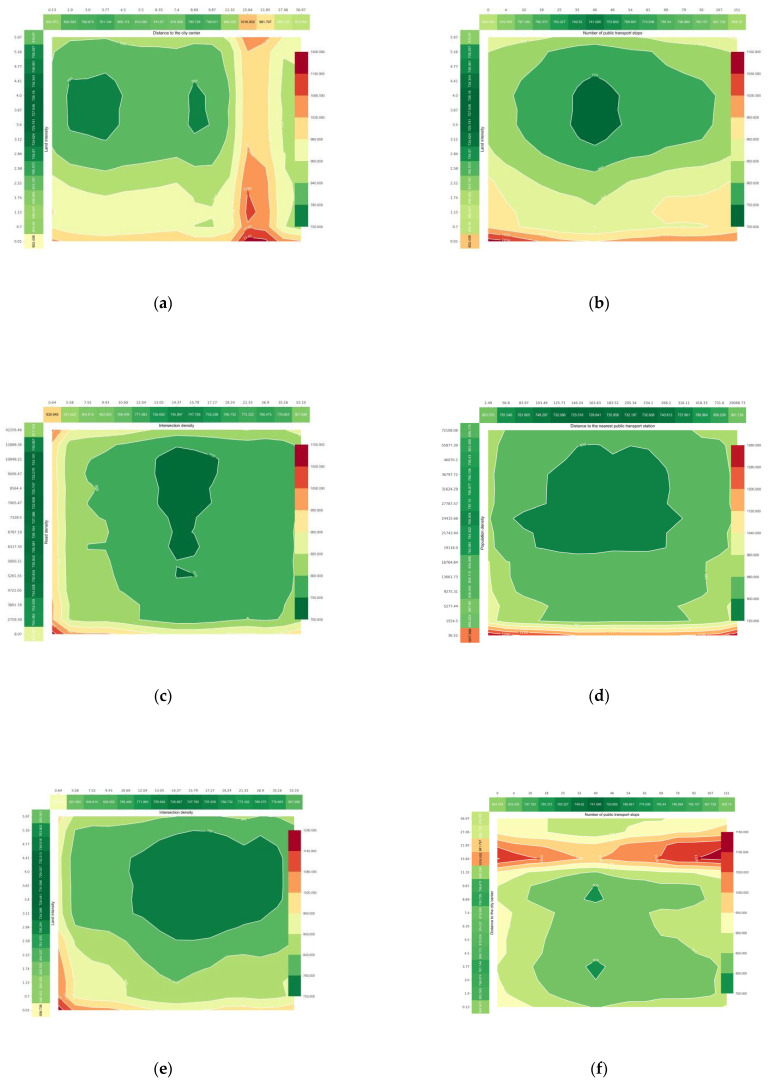
Synergistic effects among other built environment variables. (**a**) Synergy of land intensity and the distance to the city center; (**b**) Synergy of land intensity and the number of public transit stations; (**c**) Synergy of road network density and intersection density; (**d**) Synergy of population density and the distance to the nearest station; (**e**) Synergy of land intensity and intersection density; (**f**) Synergy of the distance to the city center and the number of public transit stations.

**Table 1 ijerph-20-01616-t001:** The scope and sampling of Wuhan’s previous resident travel surveys.

Year	Zone	Resident Population and Scale	Survey Population and Household Size	Sample Rate
1987	Main urban area	330,000 people 940,000 households	—	—
1998	Main urban area	3,810,000 people	76,000 people	2.0%
		150,000 households	24,000 households	
2008	City area	870,000 people	120,000 people	1.5%
		200,000 households	38,000 households	
2020	City area	12,320,000 people	40,000 people	0.5%
		4,080,000 households	15,000 households	

**Table 2 ijerph-20-01616-t002:** Carbon emission coefficient of different commuting modes.

Traffic Category	Means of Transportation	Carbon Emission Coefficient (g/(km per Person))
Small cars	Private car, unit car, car rental	135
Bus class	Bus and unit shuttle bus	50
Rail transportation category	Subway	9.1
Personal assistance class	Electric bicycle/moped, light motorcycle	8
Others	Walking, cycling	0

**Table 3 ijerph-20-01616-t003:** Variable statistics table.

Variables				Description	Mean.	Std.	Min	Max
Dependent variable	Commuting CO_2_ emissions		Daily commute CO_2_ emissions	Commuting carbon emissions per respondent per day (in grams)	717.30	1424.03	0.00	13154.26
Independent variable	Built environment	District Location	Distance from the city center	Distance to Hankou, the first-class urban center of Wuhan (in km)	10.73	9.92	0.13	56.97
			Distance to the nearest cluster center	Distance to the center of the nearest cluster (in km)	7.93	10.93	0.16	76.68
		Public transport accessibility	Distance to nearest public transport stop	Distance (in km) from the respondent’s residence to the nearest bus stop (both metro and surface bus)	0.74	3.40	0.002	29.09
			Density of public transport stations	Number of public transport stops within a 15 min walking isochronous circle of the respondent	50.25	34.56	0.00	151.00
		Density	Population density	Residential density (persons/km^2^) within a 15 min walking isochronous circle of respondents	24595.50	17363.19	36.52	72108.06
			Job density	Job density (persons/km^2^) within a 15 min walking isochronous circle of respondents	16483.62	11181.14	135.20	46206.18
			Land use intensity	Floor area ratio of sites within a 15 min walking isochronous circle of the respondent	3.05	1.46	0.01	5.97
		Design	Intersection density	Density of intersections of four or more roads within a 15 min walking isochronous circle of respondents (pcs/km^2^)	16.68	9.97	0.64	53.19
			Road network density	Density of the road network within a 15 min walking isochronous circle of the respondent (in km/km^2^)	7.28	3.58	0.01	42.26
		Diversity	Land use mixed entropy index	Mixed entropy of land use within a 15 min walking isochronous circle of respondents	0.68	0.12	0.00	0.97
			Cross River Commute	Whether the respondent has a cross-river commute, dummy variable, yes = 1, no = 0	0.07	0.25	0.00	1.00
	Socio-demographics		Age	Age of respondent	32.47	11.83	6.00	86.00
			Gender	Respondent gender, dummy variable, male = 1, female = 0	0.55	0.50	0.00	1.00
			Employment status	Whether the respondent is a full-time working employee, dummy variable, yes = 1, no = 0	0.56	0.50	0.00	1.00
			Family size	Number of family members interviewed	2.91	0.94	1.00	7.00
			Family income	Respondent’s annual household income, dummy variable, less than CNY 50,000 = 1, CNY 50,000–100,000 = 2, CNY 100,000–250,000 = 3, CNY 250,000–400,000 = 4, CNY 400,000–55,0000 = 5, CNY 550000–70,0000 = 6, greater than CNY 700,000 = 7	2.87	0.85	1.00	7.00
			Car ownership	Whether the respondent’s household owns a private car, dummy variable, yes = 1, no = 0	0.60	0.49	0.00	1.00
			Housing area	Respondent’s household housing size, dummy variable, below 40 m2 = 1, 40–70 m2 = 2, 70–90 m2 = 3, 90–110 m2 = 4, 110–120 m2 = 5, 120–150 m2 = 6, greater than 150 m2 = 7	3.59	1.11	1.00	7.00
			Education level	Respondents’ degree status, dummy variable, primary school and below = 1, middle school = 2, high school = 3, undergraduate = 4, undergraduate and above = 5, postgraduate and above = 6	3.31	1.01	1.00	6.00
	Transport Demand Management		Destination parking facilities	Does the respondent’s destination offer free parking, yes = 1, no = 0	0.18	0.39	0.00	1.00
			Transport allowance	Does the respondent have a transport subsidy, no = 0, public transport subsidy = 1, fuel, taxi subsidy = 2	0.17	0.50	0.00	2.00

**Table 4 ijerph-20-01616-t004:** Relative contribution of independent variables to commuting carbon emissions.

Variable		Rank	Importance (%)	Total Importance (%)
Built environment variable	Distance from the city center	4	6.15%	
	Distance from the nearest group center	6	5.10%	
	Distance to the nearest public transport station	10	4.46%	60.21%
	Public transportation station density	12	4.02%	
	Population density	5	5.41%	
	Job density	3	6.32%	
	Land development intensity	8	4.83%	
	Intersection density	9	4.54%	
	Mixed entropy index of land use	11	4.44%	
	Cross-river commuting	2	11.09%	
	Road network density	13	3.85%	
Socio-demographic characteristics	Age	7	4.96%	
	Gender	17	1.89%	
	Employment status	21	1.02%	
	Family size	19	1.84%	
	Family income	20	1.81%	18.75%
	Car ownership	14	3.39%	
	Housing area	16	1.99%	
	Level of education	18	1.85%	
Traffic demand management policy	Free parking	1	18.29%	
	Traffic allowance	15	2.75%	21.04%

## Data Availability

Not applicable because the used data are confidential.

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
