# Peer review of "Examining the Nonlinear and Synergistic Effects of Multidimensional Elements on Commuting Carbon Emissions: A Case Study in Wuhan, China"

_ijerph, 2023, doi:10.3390/ijerph20021616_

Round 1

Reviewer 1 Report

I appreciate the opportunity to review this study on the nonlinear and synergistic effects of multidimensional elements on commuting carbon emissions in Wuhan, China. The random forest (a machine learning technique) is used. This study addresses a valid research question, and many interesting findings are obtained. Honestly, the paper is very well written. I have no major concerns with this paper, and I admit that’s quite rare for me as a reviewer, as I rarely suggest a paper is ready to go without much need for revision. I do have a few suggested comments below, but they are relatively minor and don’t substantially change the content or communicative value of the paper.

“CO2” is not consistently used. 2 should be set as a subscript consistently.

The name of Table 1 (Basic information of data collection of four trips of residents in Wuhan) is not correct. Moreover, the paragraphs in the main body have no Table 1.

Free-parking traffic allowance has high relative importance. I suggest an in-depth discussion of the variable be performed.

The conclusions do not include any cut-off value details derived from the PDPs. A summary is needed.

The figure is of low resolution and quality. An update is recommended.

Why do you choose the random forest instead of other machine learning techniques? Some explanations are needed.

Are there theoretical and methodological implications of this study?

While I applaud the author’s attempts to raise issues that could be considered in developing new policies, some statements go too far for a research paper. Examples of some of these types of statements are as follows (others exist too):

“The population of urban centers should be controlled to avoid additional commuting CO2 emissions due to the excessive population concentration in urban centers, while the functions and population should be relieved to cluster centers with low population densities.”

“For a low-carbon land use policy, the degree of land use mixing and intensity should not be increased indefinitely.”

The wording has a few flaws as well, though I have only identified some. The paper itself, whilst understandable, needs editing by native speakers. Some errors are as follows.

complex mechanisms affecting commuting behavior.”

residential travel data.”

I recommend the author conduct a comprehensive edit of the document and suggest using a professional proofreading service when resubmitting this paper.

Considering all the comments presented here, I recommend “accepting the paper after minor revisions” in the International Journal of Environmental Research and Public Health. The paper is suitable for further consideration for the International Journal of Environmental Research and Public Health as it does demonstrate levels of significance and originality that meet the benchmark standards of excellence for this journal.

Reviewer 2 Report

This research seeks to explore the correlates of commuting carbon emissions within a non-linear analytical framework. Overall the manuscript is easy to follow, and the methods used are largely sound. However, I argue that the authors need to better clarify the raison d'etre and novelty of this research, given the extensive literature on the non-linear relationship between travel behaviour and built environment. Overall, this research fails to deliver non-trivial findings and the analyses are mostly data-driven without sufficient theoretical considerations. It is regretful, yet I have to say that the manuscript does not reach an acceptable standard at the moment.

1. Extensive studies have looked into the non-linear associations between built environment and travel behaviour (see below for examples). In this research, commuting CO2 emissions are calculated based on the multiplication between commuting distance made by different modes of transport and the corresponding emission coefficient. On this basis, the authors seek to investigate the non-linear effects of built environment on commuting CO2. While the focus on CO2 emissions is interesting, but to me, it did not add much value to the existing literature scientifically, as the findings would largely depend on the travel behaviour-built environment relation, which has been widely studied.

Liu, J., Wang, B., & Xiao, L. (2021). Non-linear associations between built environment and active travel for working and shopping: An extreme gradient boosting approach. Journal of Transport Geography, 92, 103034.

Ding, C., Cao, X. J., & Næss, P. (2018). Applying gradient boosting decision trees to examine non-linear effects of the built environment on driving distance in Oslo. Transportation Research Part A: Policy and Practice, 110, 107-117.

Cheng, L., De Vos, J., Zhao, P., Yang, M., & Witlox, F. (2020). Examining non-linear built environment effects on elderly's walking: A random forest approach. Transportation research part D: transport and environment, 88, 102552.

2. It is important to provide an explicit definition of the thermology 'synergistic effect' in the introduction. Why built environmental factors may potentially have a synergistic effect' on CO2 emissions? This should also be (briefly) clarified in the introduction from a theoretical standpoint. To me, the analyses building upon the synergistic effect is PURELY DATA-DRIVEN, and thus insights gained from such analyses and their POLICY TRANSFERABILITY would be limited.

Also, while the authors state that 'this synergistic effect is crucial for urban planners, because once certain factors (such as population density, specific age groups, and urban management policies) have been identified, they have to determine the appropriate range of indicators for other built environment factors', which seems to clarify the practical importance of considering the synergistic effect, I do not follow what the authors try to convey. An example would be useful.  

3. P2, the authors state that 'there are significant differences between China and Western countries in terms of form, built environment, and provision of facilities; therefore, the experience of Western countries may not be applicable to China', thereby trying to clarify one of the contributions of their study. Why such urban development differences can lead to a potentially different RELATIONSHIP between the built environment and commuting CO2 emissions (rather than just built environmental elements themselves)? This question needs to be answered sufficiently if focusing on the Chinese context is considered one of the contributions. It is clear that each country in the globe is different from others, but there is no need to conduct similar studies in each country, unless the rationale can be justified.

4. Please explain the reason why the City of Wuhan is a good case study area. Also, I would like to see more information regarding travel behaviour data collection (including sampling and recruitment), validation, and representativeness.

5. Please clarify how the individuals' commuting distances are calculated, given that the information the survey participant provided is trip-level start and end points. Do the authors assume that individuals travel along with the shortest OD path for calculation? If so, please critically reflect on this limitation.

6. I cannot see how the synergistic effect is modelled in the method section.
